# Research on the correlation between retinal vascular parameters and axial length in children using an AI-based fundus image analysis system

Chaoyang Zhao[1,2‡], Huilin Li[1‡], Ziyou Yuan[1], Zihan Yang[1,2], Tiantian Wang[1,2], Yan Wang[1,2], Qian Tong[3], Shaofeng Hao[1]*

1 Department of Ophthalmology, Heji Hospital Affiliated with Changzhi Medical College, Changzhi, China, 2 Graduate Office, Changzhi Medical College, Changzhi, Shanxi, China, 3 Department of Public Health and Preventive Medicine, Changzhi Medical College, Changzhi, Shanxi, China

‡ Chaoyang Zhao and Huilin Li as Co-first authors.
* hejiyanke@sina.com

## Abstract

### Objective

This study aims to utilize artificial intelligence technology to conduct an in-depth analysis of fundus data from myopic children and adolescents, thoroughly exploring the correlation between retinal vascular parameters and axial length (AL), and ultimately revealing the changing patterns of retinal vascular characteristics in children with different refractive errors. The findings aim to provide a scientific basis for the prevention, early screening, and formulation of personalized treatment strategies for myopia.

### Methods

The study selected 124 students from Jiandong Primary School in Changzhi City who underwent myopia prevention and control screening. Their axial length data were recorded, and fundus photographs were taken using the Topcon TNF506 non-mydriatic fundus camera. Subsequently, these fundus images were meticulously analyzed using the EVision AI fundus image analysis system, which is a commercial software that employs pre-trained algorithms to automatically extract retinal vascular parameters. Pearson and Spearman correlation coefficients were used to analyze the correlation between retinal vascular parameters and axial length, and multiple linear regression analysis was further conducted to explore their intrinsic associations.

### Results

The study found that in the low myopia group, axial length was significantly negatively correlated with various retinal vascular parameters, including the average diameters of arteries and veins, average vascular tortuosity, atrophy arc area, and leopard spot density. In the moderate to high myopia group, axial length also showed

**Data availability statement:** All relevant data are within the manuscript and its Supporting information files.

**Funding:** This study was funded by Major Scientific and Technological Key Project of the "Four Batches" Initiative by the Shanxi Provincial Health Commission (2022XM18). The funder supported not only the research financially but also contributed to study design, data collection, analysis, publication decisions, and manuscript preparation, as required by its funding policy.

**Competing interests:** The authors have declared that no competing interests exist.

significant negative correlations with the average diameter of arteries, some average venous tortuosity, and average vascular diameter. However, fractal dimension of vessels and average branch angle did not show significant changes across all myopia groups.

## Conclusion

This study clearly demonstrates a significant correlation between axial length and retinal vascular parameters, with notable differences in this correlation among children with different refractive errors. These findings not only provide a new perspective for understanding the pathological mechanisms of myopia but also offer important scientific evidence for the development of more precise and personalized myopia prevention and control strategies in the future. They have potential guiding significance for clinical practice and policy formulation.

---

## 1. Introduction

Myopia is a common refractive error primarily caused by an excessive axial length (Axial Length, AL) that mismatches with the refractive system, leading to the focusing of parallel light rays anterior to the retina and resulting in blurred distant vision [1,2]. In recent years, myopia has emerged as a significant global public health issue, with predictions indicating that nearly half of the world's population will be affected by 2050 [3]. Particularly alarming is the rising incidence of myopia, with a younger age of onset and a substantial increase in the prevalence of early-onset myopia among children and adolescents, further exacerbating the risk of high myopia [4]. AL plays a central role in the development and progression of myopia, with numerous studies showing that an increase in AL is one of the primary drivers of myopia onset and progression [5]. An excessively long eye axis can cause overstretching of ocular tissues such as the retina, choroid, and sclera, potentially leading to a series of ocular pathological changes [6]. Importantly, the rate of AL growth is closely related to the progression of myopia, especially in children and adolescents, where rapid AL growth is a major risk factor for the development of high myopia [7]. Patients with high myopia have an AL significantly beyond the normal range, which may induce severe complications such as retinal detachment [8], glaucoma [9], macular degeneration [10], and choroidal neovascularization [11], ultimately leading to severe visual impairment or even blindness. Therefore, monitoring and controlling AL growth is crucial for the prevention and control of myopia.

The retinal microvascular system, as the only directly observable vascular network in the human body, provides a unique perspective for studying ocular and systemic diseases [12]. By observing and quantifying morphological and functional characteristics of retinal vessels, such as vessel diameter, fractal dimension (Fractal Dimension, FD), vessel tortuosity, branch angle (Branch Angle, BA), and vascular density (Vascular Density, VD), it is possible to reveal changes in the retinal vascular network structure and provide reliable quantitative indicators for the study of various systemic

diseases. Studies have confirmed that changes in retinal vessels are closely related to systemic diseases such as cognitive impairment [13], diabetes [14], multiple sclerosis [15], and coronary heart disease [16]. With the rapid development of fundus photography technology and its analysis software, the visualization and quantitative analysis of the retinal microvascular system are increasingly becoming important tools for disease research.

Despite studies exploring the relationship between retinal vascular parameters and certain ocular and systemic diseases, research on the relationship between myopia and retinal vascular parameters remains relatively scarce, especially in children and adolescents. Currently, there is a lack of in-depth and systematic research on how retinal vascular parameters change during the development of myopia and how these changes are correlated with axial length. Therefore, further exploration of the association between retinal vascular parameters and AL not only helps to reveal the pathological mechanisms of myopia but also may provide new pathways for developing more effective prevention and treatment strategies for myopia, especially in early intervention among children and adolescents. By accurately monitoring changes in retinal vascular parameters, it is possible to detect signs of myopia onset earlier and take timely measures to control its progression.

In this context, this study utilizes artificial intelligence technology to conduct in-depth analysis of fundus data from children and adolescents with myopia, aiming to explore the correlation between retinal vascular parameters and axial length. The findings are intended to provide a scientific basis for the prevention, early screening, and personalized treatment strategies for myopia. This study not only fills a gap in the current research field but also offers new perspectives and methods for the future prevention, control, and treatment of myopia.

## 2. Materials and methods

### 2.1. General information

This study employed a cross-sectional research design, including fifth and sixth graders who underwent myopia prevention and control screening at Jiandong Primary School in Changzhi City between June 1st and June 30th, 2023. Through random sampling, 150 students were selected for screening, and ultimately, 124 eligible students were identified as the study subjects based on inclusion and exclusion criteria. Among the study subjects, there were 61 males (49.2%) and 63 females (50.8%), aged 10–12 years, with a mean age of 11.16 ± 0.63 years. Axial length (AL) ranged from 20.18 to 27.60mm, averaging 23.95 ± 1.20mm. The spherical equivalent (SE) of the right eye ranged from -8.5 to +0.38D, averaging -1.98 ± 1.76D. This study strictly adhered to the principles of the Declaration of Helsinki and received approval from the Ethics Committee of Heji Hospital Affiliated to Changzhi Medical College **(Ethical Approval Number: 202207)**.

### 2.2. Methods

#### 2.2.1. Examination items and procedures.
Inclusion Criteria: (1) Age between 10 and 12 years; (2) Able to cooperate with ophthalmic examinations; (3) Clear fundus images; (4) Students and their guardians agree to undergo refractive and fundus examinations and sign the informed consent form.

Exclusion Criteria: (1) Unable to cooperate with ophthalmic examinations; (2) presence of organic ocular diseases; (3) recent history of ocular trauma; history of previous ocular surgery; (4) students or their guardians refusing refractive or fundus examinations or not signing the informed consent form.

Data collection included students' basic information (e.g., age, gender) and comprehensive ophthalmic examination results. Ophthalmic examinations were conducted under non-mydriatic conditions using a desktop autorefractor (TOPCON RM-1). Those wearing glasses were measured after removing them. Each eye was measured three times, and the average was taken as the final result. If the difference between any two spherical power measurements was ≥ 0.50D, additional measurements were taken and averaged again. Fundus photography was performed using a high-vision Raymond TNF506 non-mydriatic fundus camera, capturing one 45° fundus color photograph centered on the macula for each eye of each subject. All image acquisitions were completed by uniformly trained and qualified technicians.

Based on the SE of the right eye, myopia was classified into three categories: normal group (0.5 D ≥ SE > -0.5 D, 27 individuals, 21.8%), low myopia group (-3.00 D < SE ≤ -0.5 D, 61 individuals, 49.2%), and moderate-to-high myopia group (SE ≤ -3.00 D, 36 individuals, 29.0%, including 32 with moderate myopia and 4 with high myopia).

**2.2.2. Analysis of retinal vascular parameters based on artificial intelligence. 2.2.2.1. Imaging process:** I. *Image Preprocessing*. Establish Region of Interest (ROI): Extract the area of interest, typically the retinal region, from the original image.

Background and Non-retinal Area Removal: Utilize methods such as threshold segmentation or edge detection to eliminate background and other non-retinal areas.

Denoising: Apply low-pass filters or other denoising techniques to remove noise from the image [17].

Normalization: Adjust the image's color, brightness, and size to ensure consistency for easier subsequent processing.

Enhancement: Use contrast-limiting adaptive histogram equalization (CLAHE) to enhance image contrast and highlight retinal features.

II. *Segmentation*. Segmentation Model Application:

Retinal Vessel Segmentation: Apply the segmentation model to the preprocessed image to identify and segment blood vessels using EVision AI [18].

Refinement of Segmentation Results: Refine segmentation results using morphological operations.

Optic Disc Segmentation: Use polar coordinate transformation and edge detection to locate the optic disc boundary on the polar coordinate image. Subsequently, perform an inverse transformation in the Cartesian coordinate system to accurately position the optic disc boundary.

III. *Parameter Extraction*. Vascular Feature Extraction: Calculate fractal dimension, average vessel diameter, tortuosity, and branching angle of the blood vessels. Extract vascular features using mathematical morphology and image processing techniques.

Quantitative Analysis: Convert the segmented vascular area into pixel-level labels for quantitative analysis. Use the box-counting method to calculate the fractal dimension of blood vessels.

Result Output: Output quantitative parameters of blood vessels, including fractal dimension, average vessel diameter, tortuosity, and branching angle.

**2.2.2.2. Image analysis:** Next, we will continue to use the artificial intelligence fundus image analysis system EVision AI [17] to conduct in-depth analysis of color fundus images centered on the macula. The analysis covered key indicators such as mean retinal artery diameter, mean retinal vein diameter, arteriovenous diameter ratio, mean artery tortuosity, mean vein tortuosity, mean vascular diameter, mean vascular tortuosity, vascular fractal dimension, mean vascular branching angle, atrophy arc area, and leopard spot density [19–22]. In addition to focusing on vascular parameters in the central macular area, we further analyzed vascular parameters in annular regions at different distances (0.5 to 1.0PD, 1.0 to 1.5PD, 1.5 to 2.0PD, 2.0 to 2.5PD) from the optic disc boundary.

For each measurement index, we provided clear definitions, such as the retinal vascular branching angle (the mean angle formed by the main vessel and branch vessels within 2.0 PD from the optic disc boundary, with larger angles potentially indicating abnormalities), vascular fractal dimension (an indicator assessing the complexity of the vascular network distribution, with higher values indicating more complex and refined distributions), and vascular tortuosity (an indicator reflecting the degree of bending or twisting of retinal vessels). The atrophy arc area represents degenerative changes in the retina or choroid, while leopard spot density reflects structural changes in the choroid and retina of the fundus, often associated with high myopia [23,24].

**2.2.2.3. Morphological parameter measurement and recording:** Finally, based on the analysis results of EVision AI, we accurately measured and recorded the morphological parameters of various fundus features. These parameters include but are not limited to vessel diameter, optic disc size, and vessel curvature, providing crucial data support for subsequent medical diagnosis and research.(Figs 1–5).

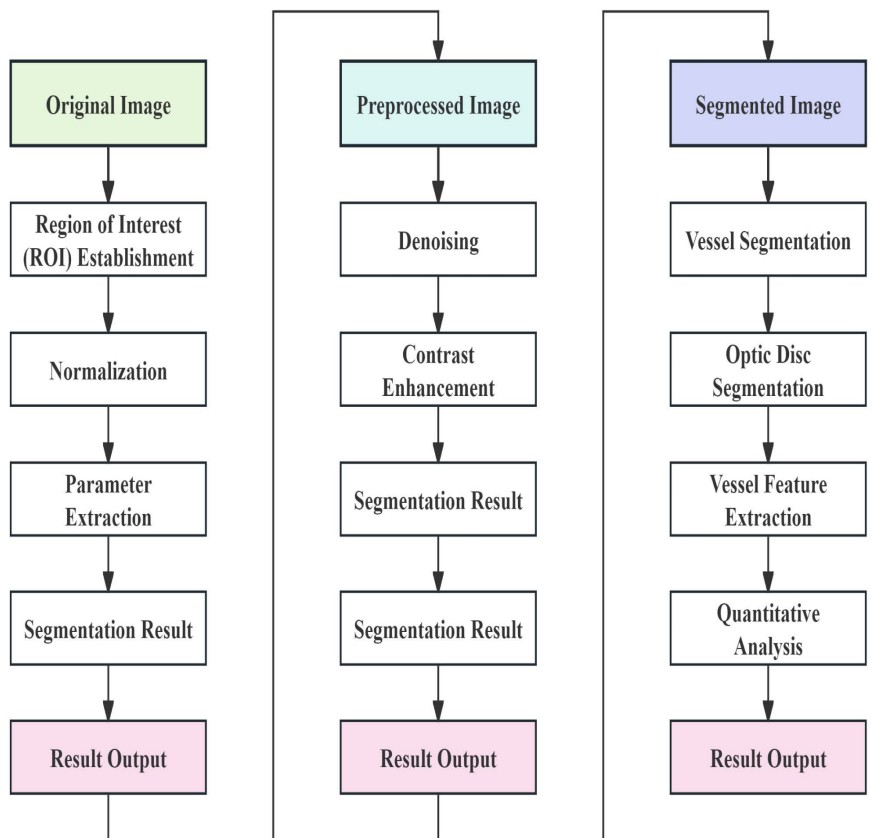

**Fig 1. Presents a flowchart of the entire imaging process and EVision AI analysis.**

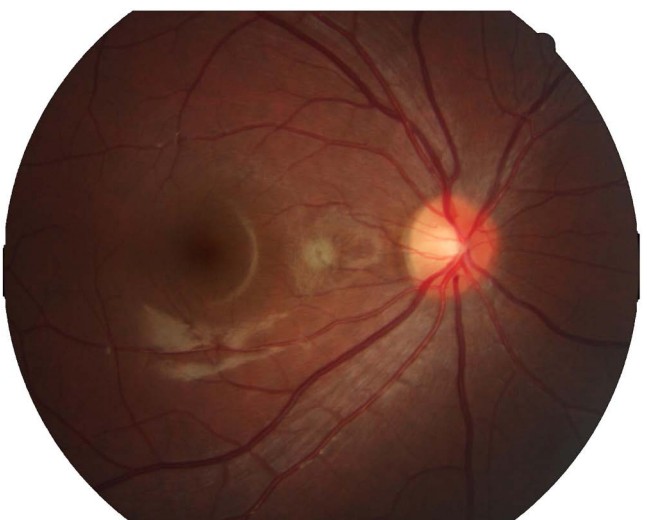

**Fig 2. Raw fundus photograph.**

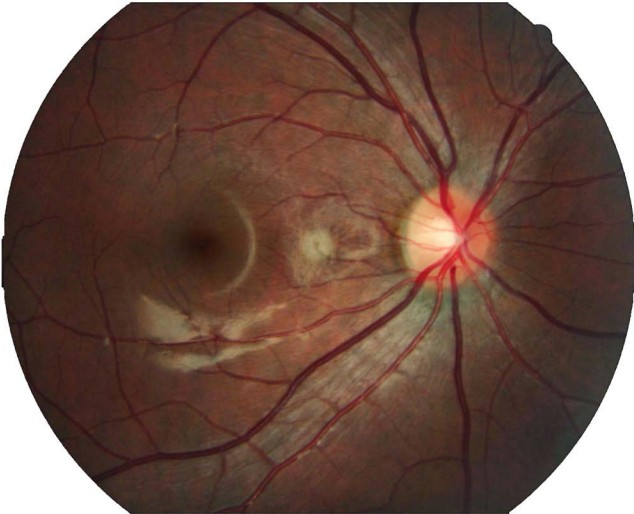

**Fig 3. Preprocessed fundus image after denoising and enhancement.**

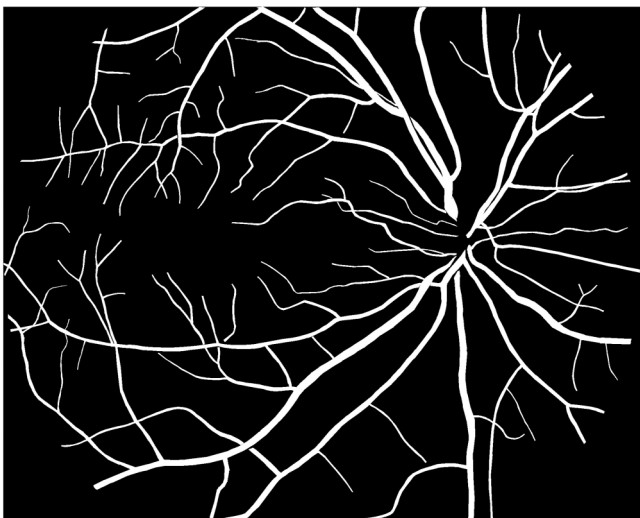

**Fig 4. Segmented retinal vessels and optic disc.**

**2.2.3. Data collection and sample size determination.** The sample size for this study was determined based on preliminary pilot experiment results and statistical power analysis, aiming to ensure sufficient statistical power to detect the expected effects. Subjects meeting the inclusion criteria were screened from fifth and sixth graders at Jiandong Primary School in Changzhi City through random sampling to ensure the representativeness and breadth of the sample.

**2.2.4. Statistical analysis methods.** Statistical analysis was conducted using SPSS 25.0 software. First, the normality of quantitative data was tested using the Shapiro-Wilk test to determine the data distribution. For normally distributed data, the mean (± standard deviation, denoted as ±S) was used for description; for non-normally distributed data, the median (interquartile range, denoted as M(IQR)) was used.

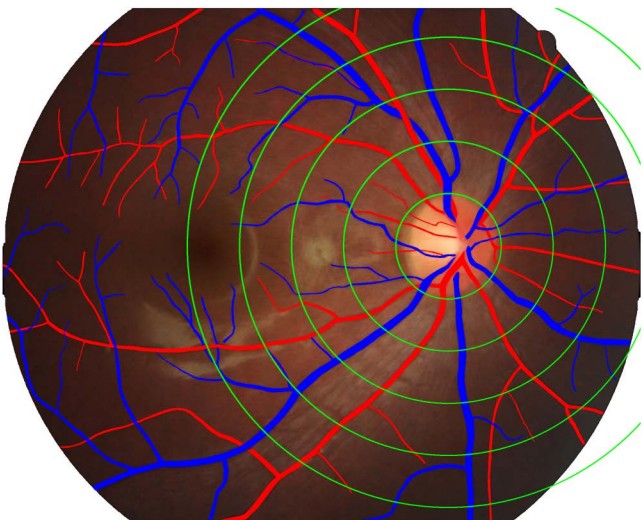

**Fig 5. Annotated arteries and veins for parameter extraction.**

For correlation analysis, based on the normality test results of the data, Pearson correlation coefficient (for normal data) and Spearman rank correlation coefficient (for non-normal data) were used to analyze the correlation between axial length (AL) and vascular parameters.

Furthermore, this study conducted multiple linear regression analysis to further explore the impact of leopard spot density and mean vascular tortuosity on axial length (AL). In this analysis, leopard spot density and mean vascular tortuosity were used as independent variables (predictors), while axial length (AL) was used as the dependent variable (response variable). The statistical significance level was set at P<0.05, indicating that results were considered statistically significant when the P-value was less than 0.05.

## 3. Results

### 3.1. Basic information analysis of subjects

Comparisons across normal, low myopia, and moderate-to-high myopia groups revealed significant differences in age, right eye spherical equivalent (SE), and axial length (AL) (P<0.05), but not in gender (P>0.05) (Table 1).

**Table 1. Comparison of Clinical Data Among Participants in the Normal, Low Myopia, and Moderate-to-High Myopia Groups.**

| Related factors | | Normal Group (n=27) | Low Myopia Group (n=61) | Moderate-to-High Myopia Group (n=36) | F/χ² | P-value |
|---|---|---|---|---|---|---|
| Age (years, `x±S) | | 10.78±0.58 | 11.25±0.57 | 11.31±0.67 | 7.15 | *0.001* |
| Gender(n, %) | Male | 13.(48.15) | 33, (54.10) | 15, (41.67) | 1.41 | 0.493 |
| | Female | 14, (51.85) | 28, (45.90) | 21, (58.33) | | |
| SE(D, `x±S) | | -0.11±0.24 | -1.51±0.74 | -4.18±1.36 | 172.63 | 0.000 |
| **Axial Length(mm, x±S)** | | **23.36±1.13** | **24.12±1.16** | **24.12±1.20** | **4.53** | *0.013* |

Note: SE - Spherical Equivalent; D - Diopter; x±S - Mean±Standard Deviation; F - F-statistic for ANOVA; χ²- Chi-square statistic for categorical variables; P-value - Significance level.

### 3.2. Univariate correlation analysis of axial length and retinal vascular parameters

**3.2.1. Axial length and retinal vascular diameters.** In low myopia, AL negatively correlated with mean artery and vein diameters (including segment-specific measurements at 1.0–1.5PD, 1.5–2.0PD, 2.0–2.5PD for arteries; 0.5–1.0PD, 2.0–2.5PD for veins) and mean vascular diameter (P<0.05). Similar correlations were observed in moderate-to-high myopia for mean artery diameters (all segments) and mean vascular diameter (P<0.05) (Table 2).

**3.2.2. Axial length and retinal vascular tortuosity.** In low myopia, AL negatively correlated with mean artery and vein tortuosity (including segment-specific values at 1.5–2.0PD) and mean vascular tortuosity (segments 0.5–1.0PD, 1.5–2.0PD) (P<0.05). In moderate-to-high myopia, only mean vein tortuosity at 0.5–1.0PD showed a significant negative correlation (P<0.05) (Table 3).

**3.2.3. Axial length and other retinal vascular parameters.** In low myopia, AL positively correlated with atrophy arc area and leopard spot density (P<0.05), while vascular fractal dimension and branching angle did not vary significantly across groups (Table 4).

### 3.3. Multiple linear regression analysis of axial length and retinal blood vessel parameters

Regression analysis indicated that a one-unit increase in tessellated fundus density correlated with a 9.743-unit increase in AL (B=9.734, P=0.013, 95% CI=2.114–17.372). Conversely, a one-unit increase in mean vessel curvature correlated with a 20,646.382-unit decrease in AL (B=-20646.382, P=0.045, 95% CI=-40777.161 to -515.602) (Table 5).

## 4. Discussion

### 4.1. Relationships between axial length and retinal vascular parameters

This study confirms a significant negative correlation between axial length (AL) and retinal vascular diameters, aligning with prior research in adults [25–27]. In low myopia, AL inversely correlated with artery and vein diameters, arteriovenous

**Table 2. Comparison of Retinal Vessel Diameters and Axial Length Among the Normal, Low Myopia, and Moderate-to-High Myopia Groups.**

| Retinal Vessel Diameter | Normal Group | | Low Myopia Group | | Moderate-to-High Myopia Group | |
|---|---|---|---|---|---|---|
| | r-value | p-value | r-value | p-value | r-value | p-value |
| Arterial Average Diameter | -0.021 | 0.918 | -0.366 | *0.004* | -0.462 | *0.005* |
| Average arterial diameter between 0.5–1.0 PD | 0.226 | 0.257 | -0.239 | 0.063 | -0.417 | *0.011* |
| Average arterial diameter between 1.0–1.5 PD | 0.085 | 0.672 | -0.292 | *0.022* | -0.349 | *0.037* |
| Average arterial diameter between 1.5–2.0 PD | 0.079 | 0.695 | -0.256 | *0.046* | -0.346 | *0.039* |
| Average arterial diameter between 2.0–2.5 PD | -0.166 | 0.408 | -0.297 | *0.020* | -0.409 | *0.013* |
| Venous Average Diameter | -0.208 | 0.298 | -0.326 | *0.010* | -0.295 | 0.081 |
| Average venous diameter between 0.5–1.0 PD | 0.080 | 0.693 | -0.290 | *0.024* | -0.096 | 0.579 |
| Average venous diameter between 1.0–1.5 PD | 0.059 | 0.769 | -0.225 | 0.081 | -0.102 | 0.554 |
| Average venous diameter between 1.5–2.0 PD | 0.032 | 0.873 | -0.043 | 0.741 | -0.179 | 0.297 |
| Average venous diameter between 2.0–2.5 PD | -0.158 | 0.431 | -0.354 | *0.005* | -0.229 | 0.179 |
| Arteriovenous Diameter Ratio | 0.197 | 0.325 | -0.159 | 0.222 | -0.124 | 0.472 |
| Arteriovenous ratio between 0.5–1.0 PD | 0.160 | 0.425 | 0.043 | 0.741 | -0.267 | 0.116 |
| Arteriovenous ratio between 1.0–1.5 PD | 0.009 | 0.964 | -0.054 | 0.677 | -0.212 | 0.215 |
| Arteriovenous ratio between 1.5–2.0 PD | -0.049 | 0.808 | -0.200 | 0.123 | -0.124 | 0.471 |
| Arteriovenous ratio between 2.0–2.5 PD | 0.008 | 0.968 | 0.031 | 0.810 | 0.045 | 0.796 |
| Average Vessel Diameter | -0.109 | 0.588 | -0.394 | *0.002* | -0.414 | *0.012* |

Note: r-value - Correlation coefficient; p-value - Significance level; PD - Papillary Diameter.

**Table 3. Comparison of Retinal Vessel Tortuosity and Axial Length Among the Normal, Low Myopia, and Moderate-to-High Myopia Groups.**

| Retinal Vessel Tortuosity | Normal Group | | Low Myopia Group | | Moderate-to-High Myopia Group | |
|---|---|---|---|---|---|---|
| | r-value | p-value | r-value | p-value | r-value | p-value |
| Arterial Average Tortuosity | -0.099 | 0.623 | -0.316 | *0.013* | 0.051 | 0.769 |
| Arterial Average Tortuosity (0.5–1.0PD) | -0.159 | 0.427 | -0.209 | 0.106 | -0.101 | 0.556 |
| Arterial Average Tortuosity (1.0–1.5PD) | -0.048 | 0.811 | -0.234 | *0.069* | -0.060 | 0.728 |
| Arterial Average Tortuosity (1.5–2.0PD) | -0.034 | 0.868 | -0.380 | *0.003* | -0.037 | 0.830 |
| Arterial Average Tortuosity (2.0–2.5PD) | -0.012 | 0.953 | -0.156 | 0.229 | -0.135 | 0.434 |
| Venous Average Tortuosity | 0.073 | 0.718 | -0.272 | 0.034 | -0.248 | 0.146 |
| Venous Average Tortuosity (0.5–1.0PD) | 0.348 | 0.075 | -0.141 | 0.277 | -0.394 | *0.017* |
| Venous Average Tortuosity (1.0–1.5PD) | -0.129 | 0.520 | -0.024 | 0.857 | -0.191 | 0.265 |
| Venous Average Tortuosity (1.5–2.0PD) | 0.100 | 0.621 | -0.292 | *0.023* | -0.136 | 0.428 |
| Venous Average Tortuosity (2.0–2.5PD) | 0.010 | 0.962 | -0.085 | 0.516 | -0.213 | 0.212 |
| Average Vessel Tortuosity | -0.020 | 0.923 | -0.343 | *0.007* | -0.181 | 0.290 |
| Average Vessel Tortuosity (0.5–1.0PD) | 0.182 | 0.363 | -0.259 | *0.044* | -0.225 | 0.186 |
| Average Vessel Tortuosity (1.0–1.5PD) | -0.102 | 0.612 | -0.225 | 0.082 | -0.123 | 0.473 |
| Average Vessel Tortuosity (1.5–2.0PD) | 0.021 | 0.915 | -0.412 | *0.001* | -0.079 | 0.647 |
| Average Vessel Tortuosity (2.0–2.5PD) | -0.005 | 0.978 | -0.195 | 0.132 | -0.236 | 0.166 |

Note: r-value - Correlation coefficient; p-value - Significance level; PD - Papillary Diameter.

**Table 4. Comparison of Other Retinal Vascular Parameters and Axial Length Among the Normal, Low Myopia, and Moderate-to-High Myopia Groups.**

| Other Parameters | Normal Group | | Low Myopia Group | | Moderate-to-High Myopia Group | |
|---|---|---|---|---|---|---|
| | r-value | p-value | r-value | p-value | r-value | p-value |
| Fractal Dimension | 0.083 | 0.681 | 0.212 | 0.101 | -0.185 | 0.280 |
| Average Branching Angle | 0.059 | 0.772 | 0.207 | 0.110 | 0.152 | 0.376 |
| Peripapillary Atrophy Area | 0.218 | 0.274 | 0.352 | *0.005* | 0.224 | 0.189 |
| Fundus Tessellation Density | 0.369 | 0.058 | 0.346 | *0.006* | -0.035 | 0.838 |

Note: r-value - Correlation coefficient; p-value - Significance level.

**Table 5. Multiple Linear Regression Analysis of Axial Length and Retinal Vascular Parameters.**

| Variable | B-value | Standard Error | Standardized Coefficient β | t-value | P-value | 95% Confidence Interval |
|---|---|---|---|---|---|---|
| Fundus Tessellation Density | 9.743 | 3.841 | 0.301 | 2.536 | *0.013* | 2.114~17.372 |
| Average Vascular Tortuosity | -20646.382 | 10135.904 | -2.108 | -2.037 | *0.045* | -40777.161~-515.602 |

Note: B-value - Regression coefficient; Standard Error - Standard error of the regression coefficient; β - Standardized regression coefficient; t-value - t-statistic; P-value - Significance level; 95% Confidence Interval - Confidence interval for the regression coefficient at the 95% level.

diameter ratio, and mean vascular diameter. This relationship likely stems from retinal thinning and stretching due to axial elongation, potentially representing a compensatory mechanism to maintain perfusion [28–31]. Retinal hypoxia, exacerbated by reduced blood flow velocity in myopia [32], may further contribute to ischemia, particularly in high myopia [33,34].

## 4.2. Vascular curvature and axial length

Abnormal vascular curvature, often linked to vascular dysfunction and blood-retina barrier damage [35,36], decreased with increasing AL in this study. This may reflect hypoxic adaptations from mechanical stretching during myopia progression [37,38], compromising retinal oxygenation and function [39].

## 4.3. Parafoveal atrophy, tessellated fundus, and axial length

AL positively correlated with parafoveal atrophy (PPA) area [21,40,41], attributed to retinal pigment epithelium and choroidal capillary loss in myopia [42–45]. Similarly, tessellated fundus density increased with AL, consistent with prior findings in Chinese adolescents [46,47], likely due to choroidal structural changes from axial elongation.

## 4.4. Fractal dimension, branching angle, and study limitations

No significant correlations were observed between AL and vascular fractal dimension or branching angle, possibly due to sample or methodological differences. Future research should explore these parameters in diverse populations.

## 4.5. Study limitations

Limitations include: (1) restricted vascular assessment from 45° macular-centered fundus images, necessitating wider-angle imaging for comprehensive evaluation; (2) cross-sectional design limiting causal inference, requiring longitudinal studies to clarify dynamic AL-vascular relationships; (3) small sample size, urging future research to include diverse age groups and myopia severities for broader generalizability; and (4) limited exploration of complex vascular parameters (e.g., fractal dimension, branching angle), warranting population-specific analyses to refine myopia prevention strategies.

## 5. Conclusion

This study demonstrates significant correlations between axial length (AL) and retinal vascular morphology in myopic children, advancing understanding of myopia pathogenesis. Leveraging AI for detailed vascular analysis, we highlight AL's impact on vessel diameters and curvature, informing personalized prevention strategies. While retinal vasculature offers insights into ocular health, future research should address imaging limitations, adopt longitudinal designs, and explore complex vascular metrics to refine myopia interventions.

## Supporting information

**S1 File. Primary Retinal Vasculature Dataset Analyzed by EVision AI.**
(XLSX)

**S2 File. Statistical Analysis Results Table.**
(XLSX)

**S3 File. Original fundus photographs (1).**
(ZIP)

**S4 File. Original fundus photographs (2).**
(ZIP)

**S5 File. The original data of vision screening.**
(XLS)

## Acknowledgments

Not applicable.

## Author contributions

**Conceptualization:** Huilin Li.

**Data curation:** Tiantian Wang, Yan Wang.

**Formal analysis:** Huilin Li, Zihan Yang.

**Funding acquisition:** Zihan Yang.

**Investigation:** Shaofeng Hao.

**Methodology:** Shaofeng Hao, Ziyou Yuan.

**Project administration:** Shaofeng Hao, Ziyou Yuan.

**Resources:** Ziyou Yuan.

**Software:** Ziyou Yuan.

**Supervision:** Qian Tong.

**Validation:** Chaoyang Zhao.

**Visualization:** Chaoyang Zhao.

**Writing – original draft:** Chaoyang Zhao.

**Writing – review & editing:** Chaoyang Zhao.

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
