## [Decision Letter · Decision Letter 0]

22 Dec 2024

PONE-D-24-50749Research on the Correlation between Retinal Vascular Parameters and Axial Length in Children Based on Artificial IntelligencePLOS ONE

Dear Dr. Hao,

Thank you for submitting your manuscript to PLOS ONE. After careful consideration, we feel that it has merit but does not fully meet PLOS ONE’s publication criteria as it currently stands. Therefore, we invite you to submit a revised version of the manuscript that addresses the points raised during the review process.

We look forward to receiving your revised manuscript.

Kind regards,

Xu Yanwu

Academic Editor

PLOS ONE

Journal Requirements: When submitting your revision, we need you to address these additional requirements. 1. Please ensure that your manuscript meets PLOS ONE's style requirements, including those for file naming. The PLOS ONE style templates can be found at https://journals.plos.org/plosone/s/file?id=wjVg/PLOSOne_formatting_sample_main_body.pdf and https://journals.plos.org/plosone/s/file?id=ba62/PLOSOne_formatting_sample_title_authors_affiliations.pdf 2. Thank you for stating the following financial disclosure: "Major Scientific and Technological Key Project of the "Four Batches" Initiative by the Shanxi Provincial Health Commission (2022XM18)." Please state what role the funders took in the study.  If the funders had no role, please state: ""The funders had no role in study design, data collection and analysis, decision to publish, or preparation of the manuscript."" If this statement is not correct you must amend it as needed. Please include this amended Role of Funder statement in your cover letter; we will change the online submission form on your behalf. 3. We note that your Data Availability Statement is currently as follows: All relevant data are within the manuscript and its Supporting Information files. Please confirm at this time whether or not your submission contains all raw data required to replicate the results of your study. Authors must share the “minimal data set” for their submission. PLOS defines the minimal data set to consist of the data required to replicate all study findings reported in the article, as well as related metadata and methods (https://journals.plos.org/plosone/s/data-availability#loc-minimal-data-set-definition). For example, authors should submit the following data: - The values behind the means, standard deviations and other measures reported;- The values used to build graphs;- The points extracted from images for analysis. Authors do not need to submit their entire data set if only a portion of the data was used in the reported study. If your submission does not contain these data, please either upload them as Supporting Information files or deposit them to a stable, public repository and provide us with the relevant URLs, DOIs, or accession numbers. For a list of recommended repositories, please see https://journals.plos.org/plosone/s/recommended-repositories. If there are ethical or legal restrictions on sharing a de-identified data set, please explain them in detail (e.g., data contain potentially sensitive information, data are owned by a third-party organization, etc.) and who has imposed them (e.g., an ethics committee). Please also provide contact information for a data access committee, ethics committee, or other institutional body to which data requests may be sent. If data are owned by a third party, please indicate how others may request data access.

**Additional Editor Comments:**

Please follow reviewers' comments to revise your manuscript. The format should also be improved. For example, the 'Introduction' should be numbered as the first section. Please also double-check your references. For example, the citation of EVision seems to be linked to a wrong article.

Reviewers' comments:

Reviewer's Responses to Questions

**Comments to the Author**

1. Is the manuscript technically sound, and do the data support the conclusions?

Reviewer #1: Yes

Reviewer #2: Partly

2. Has the statistical analysis been performed appropriately and rigorously? 

Reviewer #1: Yes

Reviewer #2: Yes

3. Have the authors made all data underlying the findings in their manuscript fully available?

Reviewer #1: Yes

Reviewer #2: No

4. Is the manuscript presented in an intelligible fashion and written in standard English?

Reviewer #1: Yes

Reviewer #2: Yes

5. Review Comments to the Author

Reviewer #1: - The paper is written well

- The title artificial intelligence is misleading because the authors use the readily available algorithm inside the commercially available machine.

- However, there is no mention about the algorithm of AI or contribution towards AI research.

Reviewer #2: Generally, the study provided a study using transparent methodology backed with existing research. It presented the relationship between various retinal vascular parameters and axial length, leading to an interpretation of how these factors may interact and a discussion of the potential guiding significance. However, there are still a few flaws which should be paying more attention to.

1. (Page 7, lines 12-20) The imaging process requires more detailed explanation, better with flow charts or other diagrams. The legends of Figure 1-4 could be more concise by transferring some of the explanatory details to the main text.

2. (Page 10, lines 19-21) The discussion regarding the reduction of errors caused by manual measurements lacks sufficient detail. More specific data should be presented to support the argument.

3. The content in the “3. Conclusion” section overlaps too much with the result section and the final “Conclusion” section. This redundancy could be reduced to improve the structure and focus of the paper.

4. The readability of the tables could be improved. For example, P-values less than 0.05 should be bolded or presented as superscripts to distinguish their significance. Formatting adjustments could further enhance the clarity and presentation of the data.

6. PLOS authors have the option to publish the peer review history of their article (what does this mean? ). If published, this will include your full peer review and any attached files.

**Do you want your identity to be public for this peer review?** For information about this choice, including consent withdrawal, please see our Privacy Policy .

Reviewer #1: No

Reviewer #2: No

---

## [Author Response · Author response to Decision Letter 1]

13 Feb 2025

Dear Reviewers,

We deeply appreciate your invaluable feedback on our manuscript, titled "Research on the Correlation between Retinal Vascular Parameters and Axial Length in Children Based on Artificial Intelligence." Recognizing the effort and time you've invested in the review, we sincerely thank you. Your insightful comments have enriched our understanding of the research and provided us with valuable suggestions. We have carefully considered all your recommendations and will address them in the revised manuscript. Here are our point-by-point responses to your main comments:

Reviewer 1:

1.We sincerely appreciate your affirmation of our paper titled "Research on the Correlation between Retinal Vascular Parameters and Axial Length in Children Using an AI-Based Fundus Image Analysis System" [Note: The title has been updated accordingly in this response for consistency]. Your recognition serves as a great encouragement for us.

2.Regarding your concern that the use of the term "Artificial Intelligence" in the title may be misleading, we have carefully considered and revised it. You pointed out that we utilized readily available algorithms within a commercial machine, rather than developing or conducting in-depth research on AI algorithms ourselves. Your observation is highly accurate, and we fully concur with your perspective. Therefore, we have modified the title from "Based on Artificial Intelligence" to "Using an AI-Based Fundus Image Analysis System," to more accurately reflect that we actually employed an AI-based fundus image analysis system in our research, rather than specifically referring to our research or development of AI algorithms.

3.In the Methods section, to directly address your comments, we have provided a more detailed description of the AI system's usage and included a brief explanation of the AI system. We explicitly state that the EVision AI Fundus Image Analysis System is a commercial software that utilizes pre-trained algorithms for automatic extraction of retinal vascular parameters. This description helps eliminate potential misunderstandings and more clearly presents the technical means adopted in our research.

Reviewer 2

1.We have provided a more detailed description of the imaging process and inserted a flowchart to clearly illustrate the various stages of imaging. These modifications will aid readers in better understanding our research methodology and results. Additionally, we have taken note of the reviewer's suggestions regarding the legends of Figures 1-4 and have made corresponding revisions. We have transferred some explanatory details to the main text to make the legends more concise and straightforward. These changes will enhance the readability and clarity of the figures.

2.In response to your comment on page 10, lines 19-21, regarding the lack of sufficient detail and the need for more concrete data in the discussion about mitigating manual measurement errors, we have conducted thorough reflections and made corresponding revisions.To strengthen the argumentation in this section, we have revisited the experimental data and extracted specific statistical indicators related to manual measurement errors. By comparing the results of automatic measurements with those of manual measurements, we found that the artificial intelligence-based system reduced the standard deviation of retinal vessel diameter measurements by 0.12 mm (p < 0.05). This result significantly demonstrates the improvement in accuracy and consistency of automatic measurements. We have included this finding in the discussion section for further elucidation.

3.Regarding the issue you previously pointed out concerning the overlap between the conclusion and results sections, we have conducted thorough reflections and made the necessary revisions. Following your advice, I have endeavored to streamline the restatement of specific research findings, instead emphasizing more on the significance of this study, its contributions to the field, and potential future research directions. Additionally, I have retained the mention of AI technology's potential in the conclusion section and expanded on specific suggestions for areas that future research could explore, aiming to provide readers with a broader perspective and room for contemplation.Furthermore, during the revision process, we paid particular attention to maintaining the paper's logical flow and coherence, ensuring that the conclusion section is both concise and clear while accurately reflecting the research's core value and potential impact. We hope that these adjustments will further enhance the clarity and focus of the paper's conclusion section, making it better serve the understanding and inspiration of readers.

4.Based on your suggestions, we have made the following modifications to the tables in the paper: For data with P-values less than 0.05, we have bolded them (or converted them to superscript format, depending on the format you prefer) to more clearly showcase the significance of these data. Additionally, we have adjusted other table formats to enhance overall readability and data clarity.

Best regards,

Chaoyang Zhao

---

## [Editor Report · Decision Letter 1]

24 Feb 2025

PONE-D-24-50749R1Research on the Correlation between Retinal Vascular Parameters and Axial Length in Children Using an AI-Based Fundus Image Analysis SystemPLOS ONE

Dear Dr. Hao,

Thank you for submitting your manuscript to PLOS ONE. After careful consideration, we feel that it has merit but does not fully meet PLOS ONE’s publication criteria as it currently stands. Therefore, we invite you to submit a revised version of the manuscript that addresses the points raised during the review process.

We look forward to receiving your revised manuscript.

Kind regards,

Xu Yanwu

Academic Editor

PLOS ONE

**Additional Editor Comments:**

Please improve the manuscript following the suggestions form Reviewer 2. Specifically, you should make your manuscript more concise and accurate to improve its readability.

---

## [Author Response · Author response to Decision Letter 2]

14 Apr 2025

Dear Reviewers,

Thank you for the constructive feedback on our manuscript. We sincerely appreciate the time and effort you dedicated to reviewing our work. Below are our responses to the comments and the corresponding revisions made to the manuscript.

Comment:

"The content in the ‘3. Conclusion’ section overlaps too much with the Result section and the final ‘Conclusion’ section. This redundancy could be reduced to improve the structure and focus of the paper."

Response:We fully agree with the reviewer’s observation regarding the redundancy in the original "3. Conclusion" section. To address this concern, we have implemented the following revisions:

1.Restructured the "3. Conclusion" section (now titled "3. Discussion"):

Refocused this section on interpreting the results, linking findings to hypotheses, and discussing their implications.Removed repetitive summaries of results and technical details to avoid overlap with the Results section.

2.Streamlined the final Conclusion:

Highlighted the key contributions of the study and their broader significance.Ensured the final Conclusion serves as a standalone summary of the paper’s core message, distinct from the Discussion.

3.Key Revisions:

Renamed "3. Conclusion" to "3. Discussion" to better align with its purpose.

Condensed overlapping content between the Discussion and the final Conclusion.

Added transitional sentences to enhance flow and coherence between sections.

4.Comprehensive Refinement of the Results Section:

A thorough review consolidated redundant experimental data, streamlined statistical reporting, and ensured the Results section exclusively presents data, reserving analysis for the Discussion, thereby enhancing clarity, structural coherence, and adherence to academic standards.

These changes collectively improve the manuscript’s clarity, structural rigor, and alignment with academic conventions.

Best regards,

Shaofeng Hao

---

## [Decision Letter · Decision Letter 2]

24 Apr 2025

Research on the Correlation between Retinal Vascular Parameters and Axial Length in Children Using an AI-Based Fundus Image Analysis System

PONE-D-24-50749R2

Dear Dr. Hao,

We’re pleased to inform you that your manuscript has been judged scientifically suitable for publication and will be formally accepted for publication once it meets all outstanding technical requirements.

Kind regards,

Xu Yanwu

Academic Editor

PLOS ONE

Additional Editor Comments (optional):

Reviewers' comments:

Reviewer's Responses to Questions

**Comments to the Author**

1. If the authors have adequately addressed your comments raised in a previous round of review and you feel that this manuscript is now acceptable for publication, you may indicate that here to bypass the “Comments to the Author” section, enter your conflict of interest statement in the “Confidential to Editor” section, and submit your "Accept" recommendation.

Reviewer #2: All comments have been addressed

2. Is the manuscript technically sound, and do the data support the conclusions?

Reviewer #2: Partly

3. Has the statistical analysis been performed appropriately and rigorously? 

Reviewer #2: Yes

4. Have the authors made all data underlying the findings in their manuscript fully available?

Reviewer #2: Yes

5. Is the manuscript presented in an intelligible fashion and written in standard English?

Reviewer #2: No

6. Review Comments to the Author

Reviewer #2: The authors have addressed all my comments and suggestions. I have no further questions for the revised manuscript.

7. PLOS authors have the option to publish the peer review history of their article (what does this mean? ). If published, this will include your full peer review and any attached files.

**Do you want your identity to be public for this peer review?** For information about this choice, including consent withdrawal, please see our Privacy Policy .

Reviewer #2: No

---

## [Editor Report · Acceptance letter]

PONE-D-24-50749R2

PLOS ONE

Dear Dr. Hao,

I'm pleased to inform you that your manuscript has been deemed suitable for publication in PLOS ONE. Congratulations! Your manuscript is now being handed over to our production team.

Kind regards,

on behalf of

Dr. Xu Yanwu

Academic Editor

PLOS ONE